# Reduction of De Novo Lipogenesis Mediates Beneficial Effects of Isoenergetic Diets on Fatty Liver: Mechanistic Insights from the MEDEA Randomized Clinical Trial

**DOI:** 10.3390/nu14102178

**Published:** 2022-05-23

**Authors:** Giuseppina Costabile, Giuseppe Della Pepa, Dominic Salamone, Delia Luongo, Daniele Naviglio, Valentina Brancato, Carlo Cavaliere, Marco Salvatore, Paola Cipriano, Marilena Vitale, Alessandra Corrado, Angela Albarosa Rivellese, Giovanni Annuzzi, Lutgarda Bozzetto

**Affiliations:** 1Department of Clinical Medicine and Surgery, Federico II University, Via Sergio Pansini 5, 80131 Naples, Italy; giuseppina.costabile@unina.it (G.C.); giuseppe.dellapepa@unina.it (G.D.P.); dominic.salamone@unina.it (D.S.); paolacipriano2006@libero.it (P.C.); marilena.vitale@unina.it (M.V.); acorrado.ac96@libero.it (A.C.); rivelles@unina.it (A.A.R.); lutgarda.bozzetto@unina.it (L.B.); 2Task Force on Microbiome Studies, University of Naples “Federico II”, 80138 Naples, Italy; 3CNR—Institute of Biostructures and Bioimaging, Naples, Via Mezzocannone 16, 80100 Naples, Italy; delia.luongo@unina.it; 4Department of Chemical Science, Federico II University, Via Cintia 21, 80126 Naples, Italy; daniele.naviglio@unina.it; 5IRCCS Synlab SDN, Via Emanuele Gianturco, 113, 80143 Naples, Italy; valentina.brancato@synlab.it (V.B.); carlo.cavaliere@synlab.it (C.C.); direzionescientifica.irccssdn@synlab.it (M.S.)

**Keywords:** fatty liver, de novo lipogenesis, SCD1, β-hydroxybutyrate, polyphenols, MUFA

## Abstract

Background: Non-alcoholic liver steatosis (NAS) results from an imbalance between hepatic lipid storage, disposal, and partitioning. A multifactorial diet high in fiber, monounsaturated fatty acids (MUFAs), n-6 and n-3 polyunsaturated fatty acids (PUFAs), polyphenols, and vitamins D, E, and C reduces NAS in people with type 2 diabetes (T2D) by 40% compared to a MUFA-rich diet. We evaluated whether dietary effects on NAS are mediated by changes in hepatic de novo lipogenesis (DNL), stearoyl-CoA desaturase (SCD1) activity, and/or β-oxidation. Methods: According to a randomized parallel group study design, 37 individuals with T2D completed an 8-week isocaloric intervention with a MUFA diet (*n* = 20) or multifactorial diet (*n* = 17). Before and after the intervention, liver fat content was evaluated by proton magnetic resonance spectroscopy, serum triglyceride fatty acid concentrations measured by gas chromatography, plasma β-hydroxybutyrate by enzymatic method, and DNL and SCD-1 activity assessed by calculating the palmitic acid/linoleic acid (C16:0/C18:2 n6) and palmitoleic acid/palmitic acid (C16:1/C16:0) ratios, respectively. Results: Compared to baseline, mean ± SD DNL significantly decreased after the multifactorial diet (2.2 ± 0.8 vs. 1.5 ± 0.5, *p* = 0.0001) but did not change after the MUFA diet (1.9 ± 1.1 vs. 1.9 ± 0.9, *p* = 0.949), with a significant difference between the two interventions (*p* = 0.004). The mean SCD-1 activity also decreased after the multifactorial diet (0.13 ± 0.05 vs. 0.10 ± 0.03; *p* = 0.001), but with no significant difference between interventions (*p* = 0.205). Fasting plasma β-hydroxybutyrate concentrations did not change significantly after the MUFA or multifactorial diet. Changes in the DNL index significantly and positively correlated with changes in liver fat (r = 0.426; *p* = 0.009). Conclusions: A diet rich in multiple beneficial dietary components (fiber, polyphenols, MUFAs, PUFAs, and other antioxidants) compared to a diet rich only in MUFAs further reduces liver fat accumulation through the inhibition of DNL. Registered under ClinicalTrials.gov no. NCT03380416.

## 1. Introduction

Non-alcoholic liver steatosis (NAS), or fatty liver disease, is the accumulation of fat in the liver through more than 5.5% of the parenchyma as assessed by 1H-MRS [1]. Fatty liver is considered the first hint of non-alcoholic fatty liver disease (NAFLD), an ominous condition encompassing several histopathological features with clinical correlates of severity ranging from liver steatosis to hepatocarcinoma. NAFLD is tied to cardiovascular disease and risk of type 2 diabetes (T2D) onset [2]. Almost all individuals with T2D have NAFLD [3]. 

Weight loss and qualitative dietary changes, mainly consisting of replacing saturated fatty acids with monounsaturated fatty acids (MUFAs) and polyunsaturated fatty acids (PUFAs), and simple sugars with complex carbohydrates and fiber, or increasing dietary components with antioxidant properties, are the dietary maneuvers shown to reduce liver fat [4]. By changing the circulating fatty acid composition, nutritional interventions may restore the imbalance between lipid storage (coming from meals, de novo lipogenesis [DNL], and fatty acid uptake), disposal (oxidation and VLDL output), and partitioning (desaturase activity) that leads to liver fat accumulation [5]. 

High-carbohydrate diets have been shown to increase fasting and postprandial DNL, whereas substituting sugar with starch reduces DNL [6]. Similarly, supplementation with the n-3 fatty acids eicosapentaenoic acid (EPA) and docosahexaenoic acid (DHA) at a dose of 4 g/day decreases both fasting and postprandial DNL [7]. There is also evidence that the degree of fat saturation is a determinant of the propensity to enter the oxidation pathway, with PUFAs and MUFAs more easily being oxidized than saturated fat [8]. 

Polyphenols are dietary components with antioxidant properties that are inversely associated with liver steatosis [9]. They may be able to counteract liver fat accumulation by activating the hepatic PPAR-α-FGF21-AMPK-PGC-1α signaling cascade that is associated with fatty acid oxidation enhancement, DNL diminution, and the recovery of mitochondrial function [10]. 

We showed that an 8-week isocaloric multifactorial diet characterized by a high content of fiber, MUFAs, n-6 and n-3 PUFAs, polyphenols, and vitamins D, E, and C resulted in a 40% decrease in the liver fat content in people with T2D [11] compared to a diet with the same amount of MUFAs, which was already shown to significantly reduce liver fat, in our previous intervention study [12]. 

In this ancillary analysis of our previous study [11], we evaluated how the multifactorial diet and MUFA diet influence the circulating fatty acid composition and whether the observed beneficial dietary effects on fatty liver were mediated by changes in indices of hepatic DNL, SCD1 activity, and/or β-oxidation.

## 2. Materials and Methods

### 2.1. Participants and Study Design

Full details of the study design and the characteristics of the participants and diets have already been published [11]. Briefly, according to a randomized, controlled, parallel group study design, patients of both sexes with T2D regularly attending the diabetes outpatient clinic of the Federico II University Hospital (Naples, Italy) and meeting the inclusion criteria (HbA1c levels ≤7.5%; body mass index (BMI) 27–35 kg/m^2^; waist circumference, men ≥102 cm and women ≥88 cm) were assigned to either a MUFA-rich diet (MUFA diet) or a multifactorial diet for 8 weeks. Participants were asked to keep their habitual physical activity unchanged during the whole study. The assigned diets were isoenergetic (i.e., maintain body weight) and similar in total fat, MUFA, carbohydrate, and protein content. Differences between the two diets were a higher amount of fiber, polyphenols, n-3 and n-6 PUFAs, and vitamins D, E, and C in the multifactorial diet, which was also characterized by a lower glycemic index. The study protocol was carried out in accordance with the Declaration of Helsinki for clinical trials and was approved by the Ethics Committee of Federico II University. All study participants provided informed consent for participation. The study was registered at ClinicalTrials.gov (NCT03380416).

### 2.2. Experimental Procedures

Anthropometrics, metabolic data, and liver fat content were assessed before and after the 8-week dietary intervention period. Blood samples were drawn after a 12 h overnight fast to measure plasma glucose, insulin, triglycerides, cholesterol, HbA1c, β-hydroxybutyrate, and the fatty acid composition of serum triglycerides. 

### 2.3. Laboratory Methods

Plasma glucose, cholesterol, and triglyceride concentrations were assayed by enzymatic colorimetric methods (Roche Diagnostics, Milan, Italy and ABX Diagnostics, Montpellier, France) on an ABX Pentra 400 (HORIBA Medical, Montpellier, France). LDL cholesterol was calculated using the Friedewald formula. The plasma insulin concentration was measured by ELISA (DIA-source ImmunoAssay S.A., Nivelles, Belgium) on a Triturus analyzer (Diagnostic Grifols S.A., Barcelona, Spain). HbA1c was measured by high performance liquid chromatography (Agilent HPLC 1200, Santa Clara, CA, USA). 

Plasma β-hydroxybutyrate concentrations were evaluated by an enzymatic endpoint method (DiaSys Diagnostic System, Holzheim, Germany) on an automated photometric analyzer (ABX-Pentra 400; Horiba Medical Kyoto, Japan). Samples were analyzed in triplicate; normal and pathological reference samples were assayed in each batch as internal quality controls. The inter- and intra-assay variation was 0.89% and 1.75%, respectively. Fatty acid proportions in the serum triglyceride fraction were evaluated by gas chromatography (GC). First, the triglyceride fraction was separated from the total serum lipids by solid phase extraction [13]. Next, the triglyceride fraction was trans-esterified to obtain fatty acid methyl esters. GC analyses were performed on a Dani GC 1000 (DANI Instruments GC 1000, DANI Instruments, Cologno Monzese, Italy) equipped with a flame ionization detector and capillary column [13].

### 2.4. Calculations 

The homeostatic model assessment of insulin resistance (HOMA-IR) was calculated using the following formula: fasting glucose (mg/dL) × fasting insulin (μU/mL)/405 [14]. The lipogenic index was calculated as the ratio between palmitic acid and linoleic acid (C16:0/C18:2n6) [15] as an indirect index of DNL. The ratio between palmitoleic acid and palmitic acid was calculated (C16:1/C16:0) [16] as an indirect index of stearoyl-Co-A desaturase (SCD-1) activity.

### 2.5. Statistical Analysis

Data are expressed as means ± SD unless otherwise stated. Within-group differences (before–after intervention) were assessed by paired sample *t*-test. The differences between the two diets were evaluated by two-way repeated measures analysis of variance (RM-ANOVA). Shapiro–Wilk Test was performed to check if variables followed a normal distribution. Bivariate associations were assessed between changes (8th week minus baseline) in DNL, metabolic parameters and liver fat by Pearson’s correlation or Spearman correlation, if data were not normally distributed, adjusting for multiple testing. A stepwise linear regression analysis was performed to assess metabolic predictors of changes in liver fat by entering the change in liver fat as a dependent variable and changes in plasma glucose, insulin, triglycerides, β-hydroxybutyrate, and lipogenic index as independent variables. A *p*-value <0.05 was considered significant. The statistical analysis was performed according to standard methods using the SPSS software V.25 (SPSS/PC).

## 3. Results

The data presented in this paper refer to the 37 participants for whom assessment of the fatty acid composition of serum triglycerides was available: 11 men and 9 women assigned to the MUFA diet, and 9 men and 8 women assigned to the multifactorial diet (Appendix A). At baseline, gender distribution, age, anthropometrics, metabolic parameters, and liver fat content were not different between the two groups (Table 1).

### 3.1. Dietary Effects on Metabolic Outcomes

After 8 weeks of dietary intervention, the reduction in liver fat was significantly greater after the multifactorial diet than the MUFA diet (−4.1% ± 4.6% vs. −1.5% ± 2.7%, respectively, *p* = 0.040; Table 1). A significant, though clinically negligible, reduction in body weight (~1 kg) and HbA1c (−0.2%) was observed after both dietary interventions without differences between groups (Table 1). These data strictly reproduce those obtained in the original intention-to-treat analysis [11].

### 3.2. Dietary Effects on the Fatty Acid Composition of Serum Triglycerides

Dietary compliance in both groups was optimal [11]. The fatty acid composition of serum triglycerides is shown in Table 2. Myristic and palmitic acids significantly decreased after the multifactorial diet, with a significant difference between the two interventions for myristic acid. In contrast, EPA, DHA, and total n-3 PUFAs increased significantly with the multifactorial diet compared to the MUFA diet, with a significant difference between the two groups (Table 2). Linoleic acid did not change significantly after either diet, whereas oleic acid increased significantly with both (Table 2).

### 3.3. Dietary Effects on de novo Lipogenesis, Desaturase Activity, and β-oxidation

At baseline, DNL index, SCD-1 activity, and fasting β-hydroxybutyrate concentrations were not significantly different between the two groups (Figure 1). The DNL index significantly decreased after the multifactorial diet (mean ± SD, 2.2 ± 0.8 vs. 1.5 ± 0.5, *p* = 0.0001) but did not change after the MUFA diet (1.9 ± 1.1 vs. 1.9 ± 0.9, *p* = 0.949), with a significant difference between the two groups (*p* = 0.004; Figure 1a). The SCD-1 activity decreased after the multifactorial diet (0.13 ± 0.05 vs. 0.10 ± 0.03, *p* = 0.001) but did not change after the MUFA diet (0.13 ± 0.06 vs. 0.12 ± 0.03, *p* = 0.121), with no difference between the groups (*p* = 0.205; Figure 1b). The fasting plasma β-hydroxybutyrate concentration did not change after either diet (MUFA diet, 0.11 ± 0.08 vs. 0.10 ± 0.05 mmol/L, *p* = 0.706; multifactorial diet, 0.11 ± 0.05 vs. 0.10 ± 0.04 mmol/L, *p* = 0.439; Figure 1c).

### 3.4. Correlation Analyses

Changes in liver fat after the dietary intervention (8th week minus baseline) directly correlated with changes in serum triglycerides myristic acid (r = 0.453, *p* = 0.040), and inversely with changes in n-3 PUFA (r = −0469, *p* = 0.024) in the whole population. Furthermore, changes in liver fat significantly correlated with changes in DNL index in the whole population (r = 0.436, *p* = 0.042) (Figure 2A), while the correlation was no more significant in the two groups considered separately (Figure 2b,c). 

A stepwise linear regression analysis performed by entering the change in liver fat as a dependent variable and changes in plasma glucose, insulin, triglycerides, β-hydroxybutyrate, and the DNL index as independent variables predicted changes in liver fat (R^2^ = 0.118), with changes in DNL being the only significant predictor of the model (β = 0.344, *p* = 0.043).

## 4. Discussion

In this study, a multifactorial diet induced different changes in the fatty acid composition of serum triglycerides, compared to a MUFA-rich diet, and significantly influenced the hepatic fatty acid metabolism by reducing DNL. Changes in DNL were significantly related to changes in liver fat content. 

No clear difference was observed for glucose control, insulin resistance, and plasma lipids between the two diets. This could be due to different factors, such as the optimal blood glucose control of patients already at baseline and the healthy profile of both diets. 

### 4.1. De Novo Lipogenesis

In our study, the multifactorial diet induced a significant 30% decrease in the lipogenic index. This reduction in DNL index is lower than that observed by drastically reducing the amount of carbohydrates in the diet. In fact, the isocaloric substitution of carbohydrates (40% vs. 4% TEI) with fat (42% vs. 72% TEI) and protein (42% vs. 72% TEI) in a 7-day intervention induced a 79% decrease in DNL evaluated by isotopic methods in people with obesity and NAFLD [17]. In our study, the proportion of carbohydrates was the same in both dietary interventions, so the finding may be explained by differences in other dietary components, such as dietary fat, polyphenols, and fiber. Previous studies have shown no difference in isotopically assessed hepatic DNL after isoenergetic diets high or low in palmitate and linoleate in healthy people [18], or hyperenergetic diets rich in saturated fatty acids or unsaturated fats in overweight/obese adults [19]. In healthy men, supplementation with n-3 fatty acids (4 g/day) induced a 30% decrease in DNL measured by standard isotopic techniques [7]. 

In a previous study, the lipogenic index decreased 40% on average in healthy volunteers when simple sugars were substituted with starch in the context of a diet very high in carbohydrates (75%) [6]. Our multifactorial diet was rich in fiber, low-glycemic index carbohydrate foods, and polyphenols. These dietary factors may have reduced substrate availability for the lipogenic pathway by inhibiting carbohydrate absorption at the intestinal level and/or through modifications in the gut microbiota composition [20]. Changes in DNL significantly and directly correlated with changes in liver fat. In a regression analysis including several metabolic factors significantly related to liver fat content and possibly determining the determination of hepatic fat accumulation (changes in fasting plasma glucose, insulin, triglycerides, β-hydroxybutyrate, and DNL), the change in DNL was the only factor significantly related to changes in liver fat. 

The reduction of DNL as a main putative mechanism of reducing liver fat is supported by the notion that lipogenesis is one of the most abnormal pathways contributing to liver fat accumulation. DNL accounts for up to 30% of triglyceride production in people with NAFLD and insulin resistance, but accounts for only 5% in healthy people [21]. The dietary impact on DNL is also clinically meaningful in terms of diabetes risk and blood glucose control as indicated by the strong correlation between saturated fat derived by DNL and hepatic insulin resistance [22].

### 4.2. Fatty Acid Composition of Serum Triglycerides

Dietary interventions induced changes in the fatty acid profile of serum triglycerides that were consistent with the changes in dietary intake. Optimal compliance with the intervention was confirmed by the significant reduction in saturated fatty acids and the increase in PUFAs observed in serum triglycerides, especially after the multifactorial diet, while oleic acid increased similarly in both diets. The changes in myristic acid and n-3 PUFAs were directly and inversely related to changes in liver fat, with the decrease in myristic acid and increase in n-3 PUFAs being significantly associated with a decrease in liver fat content. Our findings are in line with the results of a previous intervention study showing that the reduction in palmitic and myristic acid and the increase in linoleic acid in cholesterol esters was related to changes in liver fat in obese individuals eating a diet rich in saturated fatty acids or n-6 PUFAs [23]. A large observational study recently reported similar results showing that linoleic acid was inversely associated with liver fat after adjusting for confounders [24]. In line with these findings, other studies showed that patients with NAFLD are characterized by increased MUFA and decreased linoleic acid in plasma and their fatty acid profile is linked to an upregulated lipogenesis [25,26]. 

### 4.3. Stearoyl-CoA Desaturase-1 Activity

The metabolic pathway regulated by SCD1 has a potential effect on the accumulation of hepatic fat content through the repartitioning of fatty acids at the hepatic level. Palmitate entering the SCD1 pathway is derived from DNL or dietary sources [27]. In our study, the indirect index of SCD1 activity decreased mainly after the multifactorial diet. Therefore, it is plausible that changes in SCD1 activity may have contributed to the dietary effects on liver steatosis, which is in line with previous studies with larger sample sizes than ours that showed changes in the indirect index of SCD1 directly correlating with changes in liver fat [23,27]. 

### 4.4. Beta-Oxidation

Fasting concentrations of β-hydroxybutyrate did not change after dietary interventions. Previous evidence suggests that the reduction in DNL is paralleled by an increase in beta-oxidation, especially when these changes are driven by a dietary increase in PUFAs [28]. In addition, in a previous study [9], a MUFA-rich diet reduced fasting plasma β-hydroxybutyrate in people with T2D. This discordance could arise from differences in the age and gender distribution of the participants among studies, but also the pathophysiological characteristics of their liver disease. Individuals with NAFLD may have lower [29], similar [30], or higher [31] plasma β-hydroxybutyrate concentrations than individuals without NAFLD. Moreover, the relative contribution of DNL and hepatic fatty oxidation to liver steatosis could vary with an individual’s genetic and metabolic background [32]. 

### 4.5. Strengths and Limitations

A strength of the current study is the randomized controlled design that ensures high reliability of both the clinical and mechanistic results. The main limitation is the use of indirect indices of hepatic fatty acid metabolism. The use of these indices, especially for evaluating DNL, has been questioned in the context of a cross-sectional evaluation because dietary intake in observational studies is habitually not standardized [33]. Using changes in DNL and SCD1 indices made this concern less relevant for our study. The measurements were performed in the context of a standardized dietary intake with similar amounts of palmitic acid at baseline and after the intervention. On the other hand, circulating levels of linoleic acid did not change after the diets. Another limitation of our study could be represented by the lack of plasma biomarkers of dietary fiber, vitamin and polyphenol intake. Furthermore, the latter show a low bioavailability, and therefore it is difficult to define their real contribution in the context of the overall diet [34].

## 5. Conclusions

Our study shows that a diet rich in multiple beneficial dietary components, that can be considered a variant of the Mediterranean diet, although rich also in components of other healthy dietary patterns, such as polyphenols, further reduces liver fat accumulation compared to a beneficial diet only rich in MUFAs, at least in part, by inhibiting the DNL pathway. This effect is probably mediated by the synergic effects of all dietary components on substrate availability for DNL and its hormonal and genetic regulation. These results offer new pathophysiological insights for the comprehension of NAFLD insurgence and the application of new potential non-pharmacological therapeutic approaches.

## Figures and Tables

**Figure 1 nutrients-14-02178-f001:**
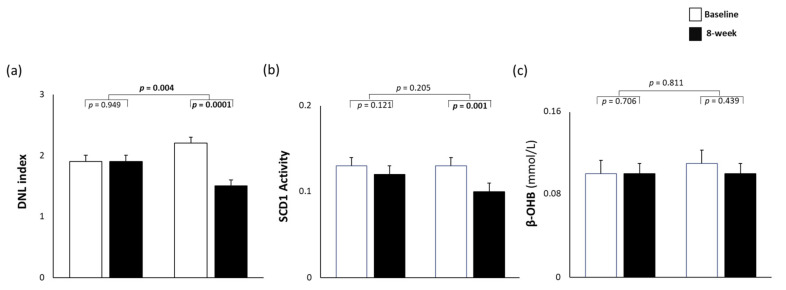
DNL index (**a**), SCD1 activity index (**b**), and plasma β-hydroxybutyrate concentration (**c**) at baseline and after the 8-week intervention with a MUFA diet (*n* = 20) or multifactorial diet (*n* = 17). MUFA, monounsaturated fatty acid; βOHB, β-hydroxybutyrate; SCD1, stearoyl-CoA desaturase. Data are presented as mean ± SEM.

**Figure 2 nutrients-14-02178-f002:**
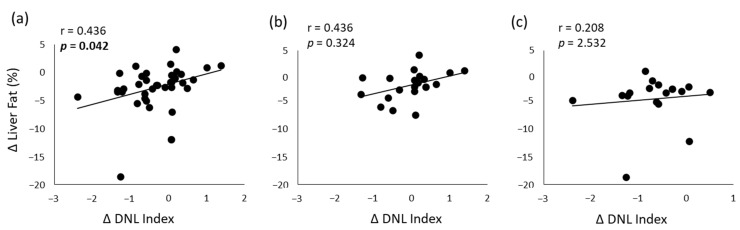
Relationships between changes (Δ: 8th week minus baseline) in the DNL index and changes in liver fat in the whole population (*n* = 37) (**a**), in the MUFA group (*n* = 20) (**b**) and in the multifactorial group (*n* = 17) (**c**). Spearman’s correlation analysis.

**Table 1 nutrients-14-02178-t001:** Anthropometrics, metabolic parameters, and liver fat content at baseline and after the 8-week intervention in participants who completed the trial.

	MUFA Diet (*n* = 20)		Multifactorial Diet (*n* = 17)		*p*-Value Diet × Time ^†^
	Baseline	8th Week	Baseline	8th Week	
Body weight (kg)	84 (15)	83 (15) *	84 (9)	83 (9) *	0.860
BMI (kg/m^2^)	31 (3)	30 (3) *	32 (4)	31 (4) *	0.535
Waist circumference (cm)	105 (10)	104 (11)	106 (10)	105 (10)	0.986
Plasma total cholesterol (mg/dL)	144 (29)	140 (26)	144 (25)	145 (32)	0.478
HDL cholesterol (mg/dL)	39 (8)	38 (8)	41 (10)	38 (8)	0.088
LDL cholesterol (mg/dL)	84 (24)	81 (21)	82 (21)	85 (26)	0.407
Plasma triglycerides (mg/dL)	109 (35)	107 (44)	105 (38)	110 (40)	0.783
Plasma glucose (mg/dL)	130 (18)	128 (17)	123 (14)	124 (19)	0.606
HbA1c (%)	6.5 (0.6)	6.3 (0.7) *	6.5 (0.4)	6.3 (0.6) *	0.681
Plasma insulin (µU/mL)	19 (10)	20 (11)	19 (9)	15 (8)	0.090
HOMA-IR	6.0 (3.0)	6.4 (3.4)	5.8 (2.8)	4.6 (2.4)	0.216
Liver fat (%)	9.9 (9.0)	8.4 (9.0) *	9.5 (7.9)	5.4 (4.9) *	0.040

Data are given as mean (SD). * *p* < 0.05 vs. baseline; ^†^ repeated-measures ANOVA. BMI: body mass index; HDL: high density lipoprotein; LDL: low density lipoprotein; HbA1c: glycated hemoglobin; HOMA-IR: homeostatic model assessment of insulin resistance.

**Table 2 nutrients-14-02178-t002:** Fatty acid composition of serum triglycerides at baseline and after the 8-week intervention in participants who completed the trial.

	MUFA Diet (*n* = 20)		Multifactorial Diet (*n* = 17)		*p*-Value Diet × Time ^†^
	Baseline	8th Week	Baseline	8th Week	
Myristic acid (%)	1.9 (0.6)	2.1 (0.9)	2.4 (0.7) ^‡^	1.8 (0.7) *	0.003
Palmitic acid (%)	21.0 (5.7)	19.2 (3.8)	21.4 (3.9)	17.6 (4.6) *	0.262
Stearic acid (%)	6.9 (3.8)	6.1 (2.3)	6.3 (2.1)	6.1 (1.6)	0.692
Palmitoleic acid (%)	2.7 (1.2)	2.2 (0.8)	2.8 (1)	1.8 (0.6) *	0.087
Oleic acid (%)	30.2 (6.9)	33.9 (6.2) *	30.8 (6.9)	34.7 (6.9) *	0.917
n-6 PUFA (%)	13.8 (3.6)	13.4 (3.4)	12.5 (3.8)	14.1(3.2)	0.121
Linoleic acid (%)	12.2 (3.5)	11.6 (4.2)	10.9 (3.9)	12.1 (3.5)	0.230
γ-Linolenic acid (%)	0.5 (0.3)	0.6 (0.5)	0.6 (0.3)	0.7 (0.5)	0.220
Arachidonic acid (%)	1.1 (0.5)	1.3 (0.6)	1.0 (0.4)	1.2 (0.5)	0.147
n-3 PUFA (%)	2.9 (1.0)	3.3 (1.1) *	2.5 (0.8)	4.4 (1.7) *	0.001
α-Linolenic acid (%)	0.7 (0.3)	0.8 (0.4)	0.7 (0.4)	1.1 (0.6) *	0.092
Eicosapentaenoic acid (%)	0.7 (0.4)	0.8 (0.5)	0.6 (0.4)	1.3 (0.8) *	0.009
Docosapentaenoic acid (%)	0.7 (0.4)	0.8 (0.5)	0.5 (0.3)	1.0 (0.8) *	0.115
Docosahexaenoic acid (%)	0.8 (0.5)	1.0 (0.6)	0.7 (0.6)	1.3 (0.9) *	0.046

Data are given as mean (SD). ^‡^
*p* < 0.05 vs. MUFA-diet at baseline; * *p* < 0.05 vs. baseline; ^†^ repeated-measures ANOVA. PUFA: polyunsaturated fatty acid.

## Data Availability

Data are available on reasonable request.

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
