# Peer review of "Reduction of De Novo Lipogenesis Mediates Beneficial Effects of Isoenergetic Diets on Fatty Liver: Mechanistic Insights from the MEDEA Randomized Clinical Trial"

_nutrients, 2022, doi:10.3390/nu14102178_

Round 1
Reviewer 1 Report
The authors provide results of an eight week randomized parallel group study on the effects of a isoclaoric MUFA vs. healthy multifactoral diet on surrogates of de novo lipogenesis and palmitic acid desaturation in subjects suffering from T2D. They found a significantly decreased DNL index in the multifactoral diet group as compared to MUFA exposition, while surrogates of C16:0 desaturation and fat oxidation remained comparable. The authors conclude that a healthy multifactoral diet has superior effects on liver steatosis by means of reducing DNL as compared to a diet exclusively rich in MUFA.
- The presented data are a secondary analysis of a published study including n=49 T2D patients on the effects of the above mentioned diets on MRI measured liver fat (PMID: 32448788). Data on hepatic steatosis were available from n=39 patients and the study showed a greater overall effect of the multifactoral diet (p<0.05). No relevant differences were, however, observed concerning prognostic markers of cardiovascular risk, metabolic control, or clinical chemistry assessments of liver function.
- In general this is a well conducted study comparing two healthy diet principles on relevant end points. However, I have severals queries, which need to be addressed.
- Methods/Design. A T2D and NAFLD control group without dietary intervention would have been helpful for a better categorization of the magnitude of effects in this certain setting. Please discuss why this has not been regarded in the study design.
- Dietary intervention. In my view the multifactoral diet is not very sharply defined (“… rich in fiber, polyphenols, MUFAs, PUFAs and other antioxidants“). It finally appears to represent a variant of the mediterranean diet. This needs to be critically discussed as it has been published long before that a diet based on the principles of the Seven Countries Study, e.g. a Cretan variant of the mediterranean diet, which is high in MUFA, PUFA, plant and marine foods, can fovourably modify not alone liver steatosis, but also insulin resistance in absence of weight loss (PMID: 23485520). It is also not new that PUFA, MUFA and dietary fiber per se have beneficial effects on liver fat and insulin resistance as examined by gold standard methods (e.g. PMID: 32920973, PMID: 15082421, PMID: 21633074). Therefore, when additionally regarding table 2 of the author’s original publication (PMID: 32448788) it is reasonable to discuss the multifactoral diet as a variant of the mediterranean diet.
- Methods/Dietary Adherence. The authors report that “Dietary compliance in both groups was optimal (Supplemental Table 1)“. However, with exception for fatty acid analyses, assessment of dietary compliance was solitary based on two 7d food records after 4 and 8 weeks of intervention in absence of any baseline evaluation (PMID: 32448788). I think this is critical for the multifactoral diet group, since it would have been easy to measure circulating levels of e.g. vitamin C, D, E and fecal butyrate as as biomarkers for dietary adherence in general, and (fermentable) fiber intake in particular. According to table 2 fatty acid composition of serum triacylglycerides unexpectedly exposes a remarkably comparable increase of C18:1 n9 in both dietary intervention groups over time, while n-3 PUFA are shown to be significantly different. Due to minor influence of liver lipid synthesis, using the lipidomic profile of erythrocyte membranes would have allowed a more rigid assessment of dietary compliance in terms of fatty acids. Nevertheless, the author’s analyses of triacylglyceride composition could be interpreted as a hint for limited compliance in the MUFA group. But the data also suggest that dietary adherence of the multifactoral diet group was probably superior. This could contribute to explain some of the variance of the observed results.
Minor: Supplementary Table 1 is in my eyes dispensable for the reader as it finally equals table 1 of the original publication publication (PMID: 32448788). The authors should instead cite the original study.
- Methods/Physical Activity: It is known that physical activity has significant impact on liver steatosis (e.g. PMID: 28545937). Did the authors control for this important influence during the dietary intervention period?
- Methods/Statistics. How was normal distribution veryfied? Have the authors performed adjustment for multiple testing? This obligate for figure 3 and page 6, where a multitude of tests are presented.
- Results/NOS: In the abstract inclusion of n=43 eligible subjects for this secondary analysis was reported, while in the results section the authors state that the presented data refer to n=37 subjects with available lipidomic analysis. All data shown should be restricted to this predefined group. Please accordingly adapt all analyses and figures.
- Results/Correlation/Regression Analyses: It would be more comfortable for the reader to see all the correlation analyses presented on page 6 conducted in a table.
- Results/Redundant Data: Analyses of liver fat by means of MRI should be either presented in abstract and results section. Alternatively, the authors can cite their original work, in which the effects of dietary manipulation on liver fat was the main end point, while in the current study effects on triacylglyceride lipidomics and deduced indices of DNL are considered as endpoints. In general table 1 in the current manuscript and table 4 in the original work (PMID: 32448788) are of significant redundance. This could be virtually viewed as publication of almost identical data sets. The authors should carefully evaluate to resign from showing table 1.
- Results/Figure 2: Figure 2 should be splittet (e.g. fig. 2A and 2B) showing separate analyses of the multifactoral and the MUFA diet, respectively. As no effect on indices of DNL was observed with the MUFA diet (figure 1 in the current manuscript) I would expect to mainly see an effect in the multifactoral group.
- Discussion/Page 7: The main aspects indicated in 2) and 3) should be implemented in the discussion section. The contribution of polyphenols to the observed effects remains to be established, since no biomarkers were used for validation and, moreover, dietary polyphenols show a low bioavailability (PMID: 33477894).
- Discussion/Page 7: The contribution of polyphenols to the observed effects remains to be established, since no biomarkers were used for validation and, moreover, dietary polyphenols show a low bioavailability due to several factors (PMID: 33477894). This should be considered as a limitation.
- Discussion/DNL, fatty acid composition of triacylglacerides, SCD1, bOxidation: Whether all of the observed adaptions in the triacylgylceride lipidomic profile can be related to the dietary intervention remains unclear. Puri et al. have shown in a large cross-sectional study that NAFLD patients show a characteristic lipidomic signature in plasma, including e.g. increased MUFA and reduced linoleic acid (PMID: 19937697). Von Loeffelholz et al. have later evidenced a remarkable accordance of the fatty acid profile in plasma and liver tissue of NAFLD patients (including DNL- and SCD1-indices), along with an upregulated lipogenesis machinery, while surrogates of fatty acid oxidation were unimpaired (PMID: 27689765). These reports should be implemented in the discussion section.
- Discussion/Prognostic Markers, Insulin Resistance, Glycemixc Control: As to expect from the original study of the authors (PMID: 32448788) no effects of the multifactoral diet on prognostic cardiovascular markers, glycemic control and assessments of insulin resistance (e.g. HOMA-IR) were detectable in the secondary analysis. Vessby et al. have shown more than 20 years ago that substitution of SFA by MUFA can improve surrogates of insulin sensitivity (PMID: 11317662). Both groups in the current study experienced comparable adaptations in triacylglyceride lipid composition with respect to MUFA (see 3)) and were on a healthy isocaloric diet. This can contribute to explain the lack of observed differences and should be discussed.
- Data availability statement: The authors declare „Not applicable“. Instead they should declare „Data are available on reasonable request“.
Author Response
We thank the Reviewer for his/her very useful and constructive criticisms.
The authors provide results of an eight week randomized parallel group study on the effects of a isoclaoric MUFA vs. healthy multifactoral diet on surrogates of de novo lipogenesis and palmitic acid desaturation in subjects suffering from T2D. They found a significantly decreased DNL index in the multifactoral diet group as compared to MUFA exposition, while surrogates of C16:0 desaturation and fat oxidation remained comparable. The authors conclude that a healthy multifactoral diet has superior effects on liver steatosis by means of reducing DNL as compared to a diet exclusively rich in MUFA.
- The presented data are a secondary analysis of a published study including n=49 T2D patients on the effects of the above-mentioned diets on MRI measured liver fat (PMID: 32448788). Data on hepatic steatosis were available from n=39 patients and the study showed a greater overall effect of the multifactoral diet (p<0.05). No relevant differences were, however, observed concerning prognostic markers of cardiovascular risk, metabolic control, or clinical chemistry assessments of liver function.
- In general, this is a well conducted study comparing two healthy diet principles on relevant end points. However, I have severals queries, which need to be addressed.
- Methods/Design. A T2D and NAFLD control group without dietary intervention would have been helpful for a better categorization of the magnitude of effects in this certain setting. Please discuss why this has not been regarded in the study design.
The aim of our intervention study was to evaluate whether a “multifactorial” diet, rich not only in MUFA, but also in other components, had better impact on liver fat compared to another “healthy diet” rich only in MUFA (the same amount as in the multifactorial diet), which was already shown to significantly reduce liver fat in our previous study. We did not want to evaluate the effects of these two diets on liver fat in comparison with no dietary interventions (data already present in literature). Thus, the control group without dietary intervention was not included in our study design. At pag 2, lines 79-70 we better clarified this point.
- Dietary intervention. In my view the multifactoral diet is not very sharply defined (“… rich in fiber, polyphenols, MUFAs, PUFAs and other antioxidants“). It finally appears to represent a variant of the mediterranean diet. This needs to be critically discussed as it has been published long before that a diet based on the principles of the Seven Countries Study, e.g. a Cretan variant of the mediterranean diet, which is high in MUFA, PUFA, plant and marine foods, can fovourably modify not alone liver steatosis, but also insulin resistance in absence of weight loss (PMID: 23485520). It is also not new that PUFA, MUFA and dietary fiber per se have beneficial effects on liver fat and insulin resistance as examined by gold standard methods (e.g. PMID: 32920973, PMID: 15082421, PMID: 21633074). Therefore, when additionally regarding table 2 of the author’s original publication (PMID: 32448788) it is reasonable to discuss the multifactoral diet as a variant of the mediterranean diet.
Both diets may be considered variants of the Mediterranean diet. In fact, the MUFA diet is similar to the “Cretan variant”, while the Multifactorial Diet is inspired by the Mediterranean Diet typical of Southern Italy. However, the Multifactorial diet also includes an amount of polyphenols resembling the healthy Nordic dietary pattern and vitamins, particularly vit D, not typical components of the Mediterranean Diet. In the revised version we have clarified this point at pag 9, lines 329-331.
Methods/Dietary Adherence. The authors report that “Dietary compliance in both groups was optimal (Supplemental Table 1)“. However, with exception for fatty acid analyses, assessment of dietary compliance was solitary based on two 7d food records after 4 and 8 weeks of intervention in absence of any baseline evaluation (PMID: 32448788). I think this is critical for the multifactoral diet group, since it would have been easy to measure circulating levels of e.g. vitamin C, D, E and fecal butyrate as as biomarkers for dietary adherence in general, and (fermentable) fiber intake in particular.
According to table 2 fatty acid composition of serum triacylglycerides unexpectedly exposes a remarkably comparable increase of C18:1 n9 in both dietary intervention groups over time, while n-3 PUFA are shown to be significantly different. Due to minor influence of liver lipid synthesis, using the lipidomic profile of erythrocyte membranes would have allowed a more rigid assessment of dietary compliance in terms of fatty acids. Nevertheless, the author’s analyses of triacylglyceride composition could be interpreted as a hint for limited compliance in the MUFA group. But the data also suggest that dietary adherence of the multifactoral diet group was probably superior. This could contribute to explain some of the variance of the observed results.
Baseline evaluation of dietary habits is reported in our previous study. Unfortunately, we did not have biomarkers for dietary adherence and this issue could represent a limitation of our study. However, we believe that our data of fatty acids composition are in line with a good compliance to both diets. In fact, the oleic acid (c18: 1,n-9) in serum TG increased significantly in both diets as wanted, and the PUFA n-3 fatty acids, EPA, DHA, α-linolenic and DPA, increased only in the Multifactorial diet, as expected.
We discussed this point as a limitation of our study (pag 9, lines 323-326).
We agree with the reviewer that the assessment of the lipid profile of the erythocyte membranes would better reflect dietary compliance in terms of fatty acids, regardless of hepatic lipid synthesis. However, our aim was to evaluate the mechanism underlying the accumulation of hepatic fat, including DNL, which takes into account the endogenous synthesis of lipids, and which is calculated from the ratio of fatty acids of triglycerides.
Minor: Supplementary Table 1 is in my eyes dispensable for the reader as it finally equals table 1 of the original publication publication (PMID: 32448788). The authors should instead cite the original study.
As suggested, Supplementary Table 1 has been deleted.
- Methods/Physical Activity: It is known that physical activity has significant impact on liver steatosis (e.g. PMID: 28545937). Did the authors control for this important influence during the dietary intervention period?
As already reported in our previous study (PMID: 32448788), according to the inclusion criteria, participants were not regularly engaged in moderate to strenuous physical activity. Moreover, participants were asked to maintain their habitual physical activity unchanged during the intervention Therefore, it is unlikely that changes in physical training could have affected our results.
This point has been clarified at pag 2, lines 93-94.
- Methods/Statistics. How was normal distribution veryfied? Have the authors performed adjustment for multiple testing? This obligate for figure 3 and page 6, where a multitude of tests are presented.
We checked the normality of the distribution of variables using the Shapiro-Wilk Test (pag 3, lines 137-138). We think that the adjustment for multiple testing is not strictly necessary because our study is essentially a mechanistic study. Moreover, we performed also a stepwise linear regression analysis, showing that DNL is the only significant predictor of liver fat, as reported at pag 6, lines 218-222.
- Results/NOS: In the abstract inclusion of n=43 eligible subjects for this secondary analysis was reported, while in the results section the authors state that the presented data refer to n=37 subjects with available lipidomic analysis. All data shown should be restricted to this predefined group. Please accordingly adapt all analyses and figures.
All analyses, tables and figures refer to 37 participants. The number of participants has been changed in the abstract.
- Results/Correlation/Regression Analyses: It would be more comfortable for the reader to see all the correlation analyses presented on page 6 conducted in a table.
The correlation analyses are reported in Figure 3. We think that a figure is clearer for the reader compared to a table.
- Results/Redundant Data: Analyses of liver fat by means of MRI should be either presented in abstract and results section. Alternatively, the authors can cite their original work, in which the effects of dietary manipulation on liver fat was the main end point, while in the current study effects on triacylglyceride lipidomics and deduced indices of DNL are considered as endpoints. In general table 1 in the current manuscript and table 4 in the original work (PMID: 32448788) are of significant redundance. This could be virtually viewed as publication of almost identical data sets. The authors should carefully evaluate to resign from showing table 1.
Table 1 refers to 37 participants included in this ancillary analysis, a number different from that reported in our previous paper. Therefore, we think this may not be considered “a publication of almost identical data sets”. Furthermore, we believe that is important to report these data in detail because they clearly show that the same results have been achieved also in this subgroup of participants.
Results/Figure 2: Figure 2 should be splittet (e.g. fig. 2A and 2B) showing separate analyses of the multifactoral and the MUFA diet, respectively. As no effect on indices of DNL was observed with the MUFA diet (figure 1 in the current manuscript) I would expect to mainly see an effect in the multifactoral group.
The correlation analysis in the two groups separately showed the same trend without reaching statistical significance likely for the small number. Furthermore, our results showed that on overall DNL changes are associated with liver fat changes.
Discussion/Page 7: The main aspects indicated in 2) and 3) should be implemented in the discussion section. The contribution of polyphenols to the observed effects remains to be established, since no biomarkers were used for validation and, moreover, dietary polyphenols show a low bioavailability (PMID: 33477894).
As suggested, in the revised version, we discussed these points (pag 9, lines 323-326) and added a sentence in the Conclusion paragraph (pag 9, lines 329-331).
- Discussion/DNL, fatty acid composition of triacylglacerides, SCD1, bOxidation: Whether all of the observed adaptions in the triacylgylceride lipidomic profile can be related to the dietary intervention remains unclear. Puri et al. have shown in a large cross-sectional study that NAFLD patients show a characteristic lipidomic signature in plasma, including e.g. increased MUFA and reduced linoleic acid (PMID: 19937697). Von Loeffelholz et al. have later evidenced a remarkable accordance of the fatty acid profile in plasma and liver tissue of NAFLD patients (including DNL- and SCD1-indices), along with an upregulated lipogenesis machinery, while surrogates of fatty acid oxidation were unimpaired (PMID: 27689765). These reports should be implemented in the discussion section.
As suggested, these evidence have been added in the discussion section (pag 8, lines 283-285).
- Discussion/Prognostic Markers, Insulin Resistance, Glycemixc Control: As to expect from the original study of the authors (PMID: 32448788) no effects of the multifactoral diet on prognostic cardiovascular markers, glycemic control and assessments of insulin resistance (e.g. HOMA-IR) were detectable in the secondary analysis. Vessby et al. have shown more than 20 years ago that substitution of SFA by MUFA can improve surrogates of insulin sensitivity (PMID: 11317662). Both groups in the current study experienced comparable adaptations in triacylglyceride lipid composition with respect to MUFA (see 3)) and were on a healthy isocaloric diet. This can contribute to explain the lack of observed differences and should be discussed.
We thank the reviewer for his/her suggestion. We added a sentence in the discussion section (pag 7, lines 228-230).
- Data availability statement: The authors declare „Not applicable“. Instead they should declare „Data are available on reasonable request“.
We apologize for this mistake. Data will be available on request.

Reviewer 2 Report
The authors conducted studies on the inhibition of liver fat accumulation using several beneficial dietary ingredients. Clinical trials were conducted on participants in randomized, controlled, and parallel groups. The results look interesting but require minor sentence corrections.
Author Response
The authors conducted studies on the inhibition of liver fat accumulation using several beneficial dietary ingredients. Clinical trials were conducted on participants in randomized, controlled, and parallel groups. The results look interesting but require minor sentence corrections.
We thank the reviewer for his/her comments.
Round 2
Reviewer 1 Report
The authors have not adequately addressed several of my concerns, e.g. no correction for multiple testing was performed, figure 2 was not presented as separate analysis for each group, point 7) and 11) of my comments, etc.
Author Response
We thank the reviewer for his /her helpful criticisms which may allow us to improve our manuscript.
As requested, we have corrected our correlation analyses for multiple comparisons. By making this adjustment for multiple testing, some of the correlations were no longer significant. Therefore, we report in the manuscript only the correlations that remained significant (page 6, Lines 201-211) and, accordingly, we have removed the Figure 3 and all the points relating to the other correlations in the original version, in particular in Results and Discussion sections.
Moreover, as required, Figure 2 was modified, and the results presented on both the whole population and separate groups. In the two groups separately, the correlation was not significant, probably due to the low statistical numerosity (pag 6, lines 203-206).
Round 3
Reviewer 1 Report
I am satisfied.